

# QuCumber: wavefunction reconstruction with neural networks

Matthew J. S. Beach[1,2], Isaac De Vlugt[2], Anna Golubeva[1,2], Patrick Huembeli[1,3], Bohdan Kulchytskyy[1,2], Xiuzhe Luo[2], Roger G. Melko[1,2*], Ejaaz Merali[2] and Giacomo Torlai[1,2,4]

**1** Perimeter Institute for Theoretical Physics, Waterloo, Ontario N2L 2Y5, Canada
**2** Department of Physics and Astronomy, University of Waterloo, Ontario N2L 3G1, Canada
**3** ICFO-Institut de Ciencies Fotoniques, Barcelona Institute of
Science and Technology, 08860 Castelldefels (Barcelona), Spain
**4** Center for Computational Quantum Physics, Flatiron Institute,
162 5th Avenue, New York, NY 10010, USA

★ rgmelko@uwaterloo.ca

## Abstract

As we enter a new era of quantum technology, it is increasingly important to develop methods to aid in the accurate preparation of quantum states for a variety of materials, matter, and devices. Computational techniques can be used to reconstruct a state from data, however the growing number of qubits demands ongoing algorithmic advances in order to keep pace with experiments. In this paper, we present an open-source software package called QuCumber that uses machine learning to reconstruct a quantum state consistent with a set of projective measurements. QuCumber uses a restricted Boltzmann machine to efficiently represent the quantum wavefunction for a large number of qubits. New measurements can be generated from the machine to obtain physical observables not easily accessible from the original data.

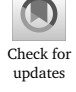

# 1 Introduction

Current advances in fabricating quantum technologies, as well as in reliable control of synthetic quantum matter, are leading to a new era of quantum hardware where highly pure quantum states are routinely prepared in laboratories. With the growing number of controlled quantum degrees of freedom, such as superconducting qubits, trapped ions, and ultracold atoms [1–4], reliable and scalable classical algorithms are required for the analysis and verification of experimentally prepared quantum states. Efficient algorithms can aid in extracting physical observables otherwise inaccessible from experimental measurements, as well as in identifying sources of noise to provide direct feedback for improving experimental hardware. However, traditional approaches for reconstructing unknown quantum states from a set of measurements, such as quantum state tomography, often suffer the exponential overhead that is typical of quantum many-body systems.

Recently, an alternative path to quantum state reconstruction was put forward, based on modern machine learning (ML) techniques [5–10]. The most common approach relies on a powerful generative model called a *restricted Boltzmann machine* (RBM) [11], a stochastic neural network with two layers of binary units. A visible layer $v$ describes the physical degrees of freedom, while a hidden layer $h$ is used to capture high-order correlations between the visible units. Given a set of neural network parameters $\lambda$, the RBM defines a probabilistic model described by the parametric distribution $p_\lambda(v)$. RBMs have been widely used in the ML community for the pre-training of deep neural networks [12], for compressing high-dimensional data into lower-dimensional representations [13], and more [14]. More recently, RBMs have been adopted by the physics community in the context of representing both classical and quantum many-body states [15,16]. They are currently being investigated for their representational power [17–19], their relationship with tensor networks and the renormalization group [20–24], and in other contexts in quantum many-body physics [25–27].

In this post, we present QuCumber: a *quantum calculator used for many-body eigenstate reconstruction*. QuCumber is an open-source Python package that implements neural-network quantum state reconstruction of many-body wavefunctions from projective measurement data. Examples of data to which QuCumber could be applied might be magnetic spin projections, orbital occupation number, polarization of photons, or the logical state of qubits. Given a training set of such measurements, QuCumber discovers the most likely compatible quantum state by finding the optimal set of parameters $\lambda$ of an RBM. A properly trained RBM is an approximation of the unknown quantum state underlying the data. It can be used to calculate various physical observables of interest, including measurements that may not be possible in the original experiment.

This post is organized as follows. In Section 2, we introduce the reconstruction technique for the case where all coefficients of the wavefunction are real and positive. We discuss the required format for input data, as well as training of the RBM and the reconstruction of typical observables. In Section 3, we consider the more general case of a complex-valued wavefunction. We illustrate a general strategy to extract the phase structure from data by performing

appropriate unitary rotations on the state before measurements. We then demonstrate a practical reconstruction of an entangled state of two qubits. Note, the detailed theory underlying the reconstruction methods used by QuCumber can be found in the original references [5, 6] and a recent review [28]. A list of useful terms and equations can be found at the end of this post in the Glossary.

## 2 Positive wavefunctions

We begin by presenting the application of QuCumber to reconstruct many-body quantum states described by wavefunctions $|\Psi\rangle$ with positive coefficients $\Psi(\mathbf{x}) = \langle \mathbf{x}|\Psi\rangle \geq 0$, where $|\mathbf{x}\rangle = |x_1, \ldots, x_N\rangle$ is a reference basis for the Hilbert space of $N$ quantum degrees of freedom. The neural-network quantum state reconstruction requires raw data $\mathcal{D} = (\mathbf{x}_1, \mathbf{x}_2, \ldots)$ generated through projective measurements of the state $|\Psi\rangle$ in the reference basis. These measurements adhere to the probability distribution given by the Born rule, $P(\mathbf{x}) = |\Psi(\mathbf{x})|^2$. Since the wavefunction is strictly positive, the quantum state is completely characterized by the measurement distribution, i.e. $\Psi(\mathbf{x}) = \sqrt{P(\mathbf{x})}$.

The positivity of the wavefunction allows a simple and natural connection between quantum states and classical probabilistic models. QuCumber employs the probability distribution $p_\lambda(\mathbf{x})$ of an RBM (see Eq. 24 of the Glossary) to approximate the distribution $P(\mathbf{x})$ underlying the measurement data. Using contrastive divergence (CD) [12], QuCumber trains the RBM to discover an optimal set of parameters $\lambda$ that minimize the Kullback-Leibler (KL) divergence between the two distributions (see Eq. 23). Upon successful training ($p_\lambda(\mathbf{x}) \sim P(\mathbf{x})$), we obtain an approximate representation of the target quantum state,

$$\psi_\lambda(\mathbf{x}) \equiv \sqrt{p_\lambda(\mathbf{x})} \simeq \Psi(\mathbf{x}). \tag{1}$$

Note, the precise mathematical form of the marginal distribution $p_\lambda(\mathbf{x})$ defined in terms of an effective energy over the parameters of the RBM is defined in the Glossary.

In the following, we demonstrate the application of QuCumber for the reconstruction of the ground-state wavefunction of the one-dimensional transverse-field Ising model (TFIM). The Hamiltonian is

$$\hat{H} = -J \sum_i \hat{\sigma}_i^z \hat{\sigma}_{i+1}^z - h \sum_i \hat{\sigma}_i^x, \tag{2}$$

where $\hat{\sigma}_i^{x/z}$ are spin-1/2 Pauli operators acting on site $i$, and we assume open boundary conditions. For this example, we consider a chain with $N = 10$ spins at the quantum critical point $J = h = 1$.

### 2.1 Setup

Given the small size of the system, the ground state $|\Psi\rangle$ can be found with exact diagonalization. The training dataset $\mathcal{D}$ is generated by sampling the distribution $P(\boldsymbol{\sigma}^z) = |\Psi(\boldsymbol{\sigma}^z)|^2$, obtaining a sequence of $N_S = 10^5$ independent spin projections in the reference basis $\mathbf{x} = \boldsymbol{\sigma}^z$.[1] Each data point in $\mathcal{D}$ consists of an array $\boldsymbol{\sigma}_j^z = (\sigma_1^z, \ldots, \sigma_N^z)$ with shape (N,) and should be passed to QuCumber as a numpy array or torch tensor. For example, $\boldsymbol{\sigma}_j^z = \mathtt{np.array([1,0,1,1,0,1,0,0,0,1])}$, where we use $\sigma_j^z = 0, 1$ to represent a spin-down and spin-up state respectively. Therefore, the entire input data set is contained in an array with shape (N_S, N).

---

[1] The training dataset can be downloaded from https://github.com/PIQuIL/QuCumber/blob/master/examples/Tutorial1_TrainPosRealWaveFunction/tfim1d_data.txt

Aside from the training data, QuCumber also allows us to import an exact wavefunction. This can be useful for monitoring the quality of the reconstruction during training. In our example, we evaluate the fidelity between the reconstructed state $\psi_\lambda(\mathbf{x})$ and the exact wavefunction $\Psi(\mathbf{x})$. The training dataset, `train_data`, and the exact ground state, `true_psi`, are loaded with the data loading utility as follows:

```
import qucumber.utils.data as data

train_path = "tfim1d_data.txt"
psi_path = "tfim1d_psi.txt"
train_data, true_psi = data.load_data(train_path, psi_path)
```

If `psi_path` is not provided, QuCumber will load only the training data.

Next, we initialize an RBM quantum state $\psi_\lambda(\mathbf{x})$ with random weights and zero biases using the constructor PositiveWaveFunction:

```
from qucumber.nn_states import PositiveWaveFunction

state = PositiveWaveFunction(num_visible=10, num_hidden=10)
```

The number of visible units (`num_visible`) must be equal to the number of physical spins $N$, while the number of hidden units (`num_hidden`) can be adjusted to systematically increase the representational power of the RBM.

The quality of the reconstruction will depend on the structure underlying the specific quantum state and the ratio of visible to hidden units, $\alpha = $ num_hidden/num_visible. In practice, we find that $\alpha = 1$ often leads to good approximations of positive wavefunctions [6]. However, in the general case, the value of $\alpha$ required for a given wavefunction should be explored and adjusted by the user.

## 2.2 Training

Once an appropriate representation of the quantum state has been defined, QuCumber trains the RBM through the function PositiveWaveFunction.fit. Several input parameters need to be provided aside from the training dataset (`train_data`). These include the number of training iterations (`epochs`), the number of samples used for the positive/negative phase of CD (`pos_batch_size`/`neg_batch_size`), the learning rate (`lr`) and the number of sampling steps in the negative phase of CD (`k`). The last argument (`callbacks`) allows the user to pass a set of additional functions to be evaluated during training.

As an example of a callback, we show the MetricEvaluator, which evaluates a function `log_every` epochs during training. Given the small system size and the knowledge of the true ground state, we can evaluate the fidelity between the RBM state and the true ground-state wavefunction (`true_psi`). Similarly, we can calculate the KL divergence between the RBM distribution $p_\lambda(\mathbf{x})$, and the data distribution $P(\mathbf{x})$, which should approach zero for a properly trained RBM. For the current example, we monitor the fidelity and KL divergence (defined in qucumber.utils.training_statistics):

```
from qucumber.callbacks import MetricEvaluator
import qucumber.utils.training_statistics as ts

log_every = 10
space = state.generate_hilbert_space(10)
callbacks = [
    MetricEvaluator(
        log_every,
        {"Fidelity": ts.fidelity, "KL": ts.KL},
```

```
        target_psi=true_psi,
        space=space,
        verbose=True,
    )
]
```

With `verbose=True`, the program will print the epoch number and all callbacks every `log_every` epochs. For the current example, we monitor the fidelity and KL divergence (see Glossary). Note that the KL divergence is only tractable for small systems. The `MetricEvaluator` will compute the KL exactly when provided with a list of all states in the Hilbert space. For convenience these can be generated with `space = state.generate_hilbert_space(10)`.

Now that the metrics to monitor during training have been chosen, we can invoke the optimization with the `fit` function of `PositiveWaveFunction`.

```
state.fit(
    train_data,
    epochs=500,
    pos_batch_size=100,
    neg_batch_size=100,
    lr=0.01,
    k=5,
    callbacks=callbacks,
)
```

Figure 1 shows the convergence of the fidelity and KL divergence during training. The convergence time will, in general, depend on the choice of hyperparameters. Finally, the network parameters $\lambda$, together with the `MetricEvaluator`'s data, can be saved (or loaded) to a file:

```
state.save(
    "filename.pt",
    metadata={
        "fidelity": callbacks[0].Fidelity,
        "KL": callbacks[0].KL,
    },
)
state.load("filename.pt")
```

With this we have demonstrated the most basic aspects of QuCumber regarding training a model and verifying its accuracy. We note that in this example the evaluation utilized the knowledge of the exact ground state and the calculation of the KL divergence, which we reemphasize is tractable only for small system sizes. However, we point out that QuCumber is capable of carrying out the reconstruction of much larger systems. In such cases, users must rely on other estimators to evaluate the training, such as expectation values of physical observables (magnetization, energy, etc). In the following, we show how to compute diagonal and off-diagonal observables in QuCumber.

## 2.3 Reconstruction of physical observables

In this section, we discuss how to calculate the average value of a generic physical observable $\hat{O}$ from a trained RBM. We start with the case of observables that are diagonal in the reference basis where the RBM was trained. We then discuss the more general cases of off-diagonal observables and entanglement entropies.

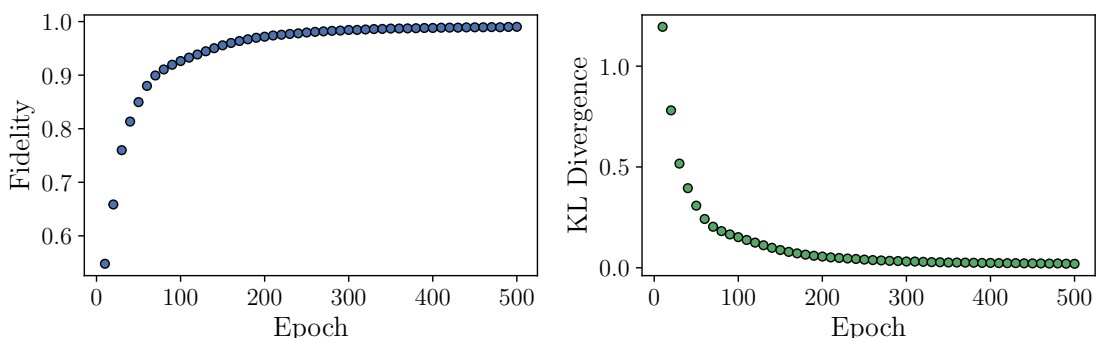

Figure 1: The fidelity (left) and the KL divergence (right) during training for the reconstruction of the ground state of the one-dimensional TFIM.

### 2.3.1 Diagonal observables

We begin by considering an observable with only diagonal matrix elements, $\langle \boldsymbol{\sigma} | \hat{\mathcal{O}} | \boldsymbol{\sigma}' \rangle = \mathcal{O}_{\boldsymbol{\sigma}} \delta_{\boldsymbol{\sigma}\boldsymbol{\sigma}'}$ where for convenience we denote the reference basis $\mathbf{x} = \boldsymbol{\sigma}^z$ as $\boldsymbol{\sigma}$ unless otherwise stated. The expectation value of $\hat{\mathcal{O}}$ is given by

$$\langle \hat{\mathcal{O}} \rangle = \frac{1}{\sum_{\boldsymbol{\sigma}} |\psi_{\lambda}(\boldsymbol{\sigma})|^2} \sum_{\boldsymbol{\sigma}} \mathcal{O}_{\boldsymbol{\sigma}} |\psi_{\lambda}(\boldsymbol{\sigma})|^2 . \tag{3}$$

The expectation value can be approximated by a Monte Carlo estimator,

$$\langle \hat{\mathcal{O}} \rangle \approx \frac{1}{N_{\text{MC}}} \sum_{k=1}^{N_{\text{MC}}} \mathcal{O}_{\boldsymbol{\sigma}_k} , \tag{4}$$

where the spin configurations $\boldsymbol{\sigma}_k$ are sampled from the RBM distribution $p_{\lambda}(\boldsymbol{\sigma})$. This process is particularly efficient given the bipartite structure of the network which allows the use of block Gibbs sampling.

A simple example for the TFIM is the average longitudinal magnetization per spin, $\langle \hat{\sigma}^z \rangle = \sum_j \langle \hat{\sigma}^z_j \rangle / N$, which can be calculated directly on the spin configuration sampled by the RBM (i.e., the state of the visible layer). The visible samples are obtained with the `sample` function of the RBM state object:

```
samples = state.sample(num_samples=1000, k=10)
```

which takes the total number of samples (`num_samples`) and the number of block Gibbs steps (`k`) as input. Once these samples are obtained, the magnetization can be calculated simply as

```
magnetization = samples.mul(2.0).sub(1.0).mean()
```

where we converted the binary samples of the RBM back into ±1 spins before taking the mean.

### 2.3.2 Off-diagonal observables

We turn now to the case of off-diagonal observables, where the expectation value assumes the following form

$$\langle \hat{\mathcal{O}} \rangle = \frac{1}{\sum_{\boldsymbol{\sigma}} |\psi_{\lambda}(\boldsymbol{\sigma})|^2} \sum_{\boldsymbol{\sigma}\boldsymbol{\sigma}'} \psi^*_{\lambda}(\boldsymbol{\sigma}) \psi_{\lambda}(\boldsymbol{\sigma}') \mathcal{O}_{\boldsymbol{\sigma}\boldsymbol{\sigma}'} . \tag{5}$$

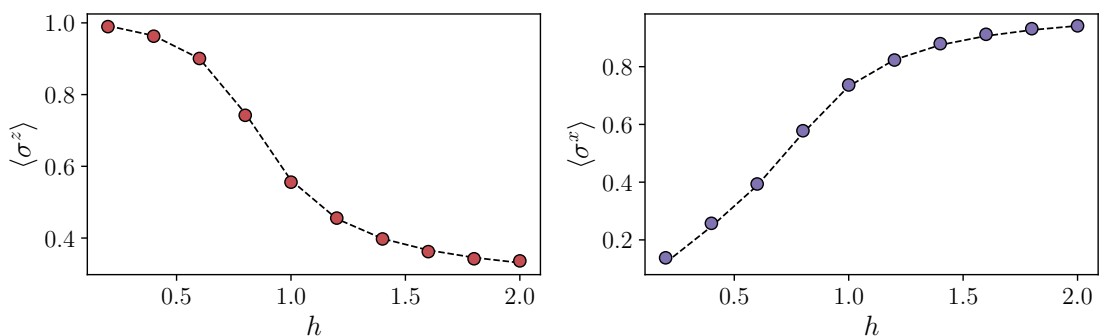

Figure 2: Reconstruction of the magnetic observables for the TFIM chain with $N = 10$ spins. We show the average longitudinal (left) and transverse (right) magnetization per site obtained by sampling from a trained RBM. The dashed line denotes the results from exact diagonalization.

This expression can once again be approximated with a Monte Carlo estimator,

$$\langle \hat{\mathcal{O}} \rangle \approx \frac{1}{N_{\mathrm{MC}}} \sum_{k=1}^{N_{\mathrm{MC}}} \mathcal{O}_{\boldsymbol{\sigma}_k}^{[L]}, \tag{6}$$

of the so-called *local estimator* of the observable:

$$\mathcal{O}_{\boldsymbol{\sigma}_k}^{[L]} = \sum_{\boldsymbol{\sigma}'} \frac{\psi_{\boldsymbol{\lambda}}(\boldsymbol{\sigma}')}{\psi_{\boldsymbol{\lambda}}(\boldsymbol{\sigma}_k)} \mathcal{O}_{\boldsymbol{\sigma}_k \boldsymbol{\sigma}'} . \tag{7}$$

As long as the matrix representation $\mathcal{O}_{\boldsymbol{\sigma}\boldsymbol{\sigma}'}$ is sufficiently sparse in the reference basis, the summation can be evaluated efficiently.

As an example, we consider the specific case of the transverse magnetization for the $j$-th spin, $\langle \hat{\sigma}_j^x \rangle$, with matrix elements

$$\langle \boldsymbol{\sigma} | \hat{\sigma}_j^x | \boldsymbol{\sigma}' \rangle = \delta_{\sigma_j', 1 - \sigma_j} \prod_{i \neq j} \delta_{\sigma_i', \sigma_j} . \tag{8}$$

Therefore, the expectation values reduces to the Monte Carlo average of the local observable

$$(\sigma_j^x)^{[L]} = \frac{\psi_{\boldsymbol{\lambda}}(\sigma_1, \ldots, 1 - \sigma_j, \ldots, \sigma_N)}{\psi_{\boldsymbol{\lambda}}(\sigma_1, \ldots, \sigma_j, \ldots, \sigma_N)} . \tag{9}$$

evaluated on spin configurations $\boldsymbol{\sigma}_k$ sampled from the RBM distribution $p_{\boldsymbol{\lambda}}(\boldsymbol{\sigma})$.

QuCumber provides an interface for sampling off-diagonal observables in the `ObservableBase` class. Thorough examples are available in the tutorial section in the documentation.[2] As an example, $\sigma^x$ can be written as an observable class with

```
import torch
from qucumber.utils import cplx
from qucumber.observables import ObservableBase

class SigmaX(ObservableBase):
    def apply(self, nn_state, samples):
```

---

[2]The observables tutorial is available at https://qucumber.readthedocs.io/en/stable/_examples/Tutorial3_DataGeneration_CalculateObservables/tutorial_sampling_observables.html

```
        psi = nn_state.psi(samples)
        psi_ratio_sum = torch.zeros_like(psi)

        for i in range(samples.shape[-1]):  # sum over sites
            flip_spin(i, samples)  # flip the spin at site i
            # add ratio psi_(-i) / psi to the running sum
            psi_flip = nn_state.psi(samples)
            psi_ratio = cplx.elementwise_division(psi_flip, psi)
            psi_ratio_sum.add_(psi_ratio)
            flip_spin(i, samples)  # flip it back

        # take real part and divide by number of spins
        return psi_ratio_sum[0].div_(samples.shape[-1])
```

The value of the observable can be estimated from a set of samples with:

```
SigmaX().statistics_from_samples(state, samples)
```

which produces a dictionary containing the mean, variance, and standard error of the observable. Similarly, the user can define other observables like the energy.

The reconstruction of two magnetic observables for the TFIM is shown in Fig. 2, where a different RBM was trained for each value of the transverse field $h$. In the left plot, we show the average longitudinal magnetization per site, which can be calculated directly from the configurations sampled by the RBM. In the right plot, we show the off-diagonal observable of transverse magnetization. In both cases, QuCumber successfully discovers an optimal set of parameters $\lambda$ that accurately approximate the ground-state wavefunction underlying the data.

### 2.3.3 Entanglement entropy

A quantity of significant interest in quantum many-body systems is the degree of entanglement between a subregion $A$ and its complement $\bar{A}$. Numerically, measurement of bipartite entanglement entropy is commonly accessed through the computation of the second Rényi entropy $S_2 = -\ln \mathrm{Tr}(\rho_A^2)$. When one has access to a pure state wavefunction $\psi_\lambda(\mathbf{x})$, Rényi entropies can be calculated as an expectation value of the "Swap" operator [29],

$$S_2 = -\ln \left\langle \widehat{\mathrm{Swap}_A} \right\rangle. \tag{10}$$

It is essentially an off-diagonal observable that acts on an extended product space consisting of two independent copies of the wavefunction, $\psi_\lambda(\mathbf{x}) \otimes \psi_\lambda(\mathbf{x})$, referred to as "replicas". As the name suggests, the action of the Swap operator is to swap the spin configurations in region $A$ between the replicas,

$$\widehat{\mathrm{Swap}_A} |\boldsymbol{\sigma}_A, \boldsymbol{\sigma}_{\bar{A}}\rangle_1 \otimes |\boldsymbol{\sigma}'_A, \boldsymbol{\sigma}'_{\bar{A}}\rangle_2 = |\boldsymbol{\sigma}'_A, \boldsymbol{\sigma}_{\bar{A}}\rangle_1 \otimes |\boldsymbol{\sigma}_A, \boldsymbol{\sigma}'_{\bar{A}}\rangle_2. \tag{11}$$

Here the subscript of the ket indicates the replica index, while the two labels inside a ket, such as $\boldsymbol{\sigma}_A, \boldsymbol{\sigma}_{\bar{A}}$, describe the spins configurations within the subregion and its complement.

In QuCumber, the Swap operator is implemented as a routine within the `entanglement` observable unit,

```
def swap(s1, s2, A):
    _s = s1[:, A].clone()
    s1[:, A] = s2[:, A]
    s2[:, A] = _s
    return s1, s2
```

where s1 and s2 are batches of samples produced from each replica, and A is a list containing the indices of the sites in subregion *A*. While ideally those samples should be entirely independent, in order to save computational costs, QuCumber just splits a given batch into two equal parts and treats them as if they were independent samples. This is implemented within the SWAP observable,

```python
class SWAP(ObservableBase):
    def __init__(self, A):
        self.A = A

    def apply(self, nn_state, samples):
        _ns = samples.shape[0] // 2
        samples1 = samples[:_ns, :]
        samples2 = samples[_ns : _ns * 2, :]

        psi_ket1 = nn_state.psi(samples1)
        psi_ket2 = nn_state.psi(samples2)

        psi_ket = cplx.elementwise_mult(psi_ket1, psi_ket2)
        psi_ket_star = cplx.conjugate(psi_ket)

        samples1_, samples2_ = swap(
            samples1, samples2, self.A
        )
        psi_bra1 = nn_state.psi(samples1_)
        psi_bra2 = nn_state.psi(samples2_)

        psi_bra = cplx.elementwise_mult(psi_bra1, psi_bra2)
        psi_bra_star = cplx.conjugate(psi_bra)

        return cplx.real(
            cplx.elementwise_division(
                psi_bra_star, psi_ket_star
            )
        )
```

Note the similarity in the implementation to that for the transverse magnetization observable from the last section, once the amplitude of a sample is substituted with the product of amplitudes drawn from each replica.

Using this observable, we can estimate the Rényi entropy of the region containing the first 5 sites in the chain using Eq. 10,

```python
A = [0, 1, 2, 3, 4]
swap_ = SWAP(A)
swap_stats = swap_.statistics_from_samples(state, samples)
S_2 = -np.log(swap_stats["mean"])
```

We apply this measurement procedure to a TFIM chain with results shown in Fig. 3. As was the case with the magnetization observables, the trained RBM gives a good approximation to the second Rényi entropy for different subregion *A* sizes. Being a basis-independent observable, this constitutes a useful test on the ability of QuCumber to capture the full wavefunction from the information contained in a single-basis dataset for TFIM.

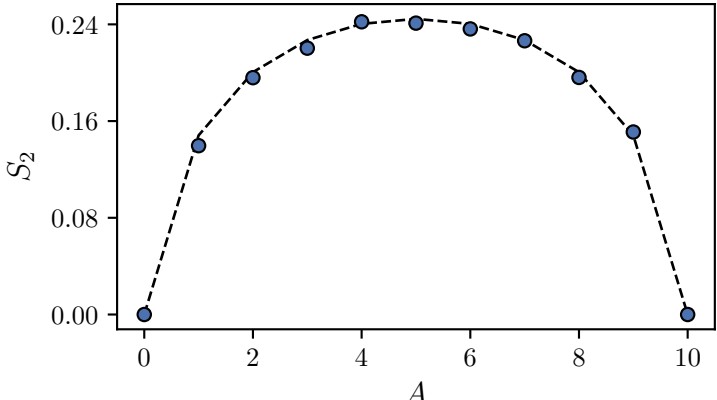

Figure 3: The second Rényi entropy for the TFIM chain with $N = 10$ spins. The number of sites in the entangled bipartition $A$ is indicated by the horizontal axis. The markers indicate values obtained through the "Swap" operator applied to the samples from a trained RBM. The dashed line denotes the result from exact diagonalization.

## 3  Complex wavefunctions

For positive wavefunctions, the probability distribution underlying the outcomes of projective measurements in the reference basis contains all possible information about the unknown quantum state. However, in the more general case of a wavefunction with a non-trivial sign or phase structure, this is not the case. In this section, we consider a target quantum state where the wavefunction coefficients in the reference basis can be complex-valued, $\Psi(\boldsymbol{\sigma}) = \Phi(\boldsymbol{\sigma})e^{i\theta(\boldsymbol{\sigma})}$. We continue to choose the reference basis as $\boldsymbol{\sigma} = \boldsymbol{\sigma}^z$. We first need to generalize the RBM representation of the quantum state to capture a generic complex wavefunction. To this end, we introduce an additional RBM with marginalized distribution $p_\mu(\boldsymbol{\sigma})$ parameterized by a new set of network weights and biases $\boldsymbol{\mu}$. We use this to define the quantum state as:

$$\psi_{\lambda\mu}(\boldsymbol{\sigma}) = \sqrt{p_\lambda(\boldsymbol{\sigma})}e^{i\phi_\mu(\boldsymbol{\sigma})/2}, \tag{12}$$

where $\phi_\mu(\boldsymbol{\sigma}) = \log(p_\mu(\boldsymbol{\sigma}))$ [6]. In this case, the reconstruction requires a different type of measurement setting. It is easy to see that projective measurements in the reference basis do not convey any information on the phases $\theta(\boldsymbol{\sigma})$, since $P(\boldsymbol{\sigma}) = |\Psi(\boldsymbol{\sigma})|^2 = \Phi^2(\boldsymbol{\sigma})$.

The general strategy to learn a phase structure is to apply a unitary transformation $\mathcal{U}$ to the state $|\Psi\rangle$ before the measurements, such that the resulting measurement distribution $P'(\boldsymbol{\sigma}) = |\Psi'(\boldsymbol{\sigma})|^2$ of the rotated state $\Psi'(\boldsymbol{\sigma}) = \langle\boldsymbol{\sigma}|\,\mathcal{U}\,|\Psi\rangle$ contains fingerprints of the phases $\theta(\boldsymbol{\sigma})$ (Fig. 4). In general, different rotations must be independently applied to gain full information on the phase structure. We make the assumption of a tensor product structure of the rotations, $\mathcal{U} = \bigotimes_{j=1}^{N} \hat{\mathcal{U}}_j$. This is equivalent to a local change of basis from $|\boldsymbol{\sigma}\rangle$ to $\{|\boldsymbol{\sigma}^b\rangle = |\sigma_1^{b_1}, \dots, \sigma_N^{b_N}\rangle\}$, where the vector $\boldsymbol{b}$ identifies the local basis $b_j$ for each site $j$. The target wavefunction in the new basis is given by

$$\Psi(\boldsymbol{\sigma}^b) = \langle\boldsymbol{\sigma}^b|\Psi\rangle = \sum_{\boldsymbol{\sigma}}\langle\boldsymbol{\sigma}^b|\boldsymbol{\sigma}\rangle\langle\boldsymbol{\sigma}|\Psi\rangle = \sum_{\boldsymbol{\sigma}}\mathcal{U}(\boldsymbol{\sigma}^b, \boldsymbol{\sigma})\Psi(\boldsymbol{\sigma}), \tag{13}$$

and the resulting measurement distribution is

$$P_b(\boldsymbol{\sigma}^b) = \left|\sum_{\boldsymbol{\sigma}}\mathcal{U}(\boldsymbol{\sigma}^b, \boldsymbol{\sigma})\Psi(\boldsymbol{\sigma})\right|^2. \tag{14}$$

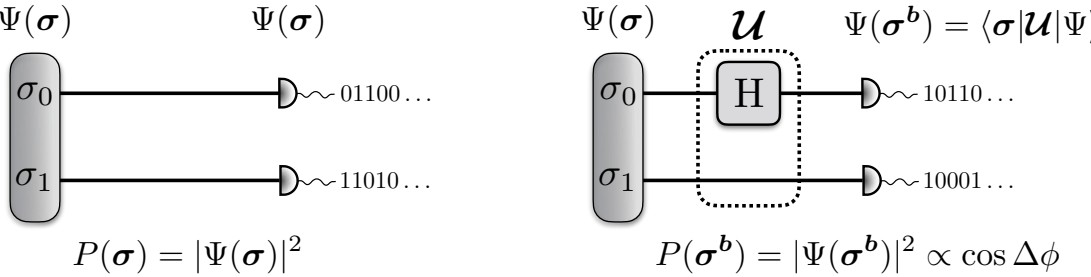

Figure 4: Unitary rotations for two qubits. (left) Measurements on the reference basis. (right) Measurement in the rotated basis. The unitary rotation (the Hadamard gate on qubit $\sigma_0$) is applied after state preparation and before the projective measurement.

To clarify the procedure, let us consider the simple example of a quantum state of two qubits:

$$|\Psi\rangle = \sum_{\sigma_0,\sigma_1} \Phi_{\sigma_0\sigma_1} e^{i\theta_{\sigma_0\sigma_1}} |\sigma_0\sigma_1\rangle, \tag{15}$$

and rotation $\mathcal{U} = \hat{H}_0 \otimes \hat{\mathcal{I}}_1$, where $\hat{\mathcal{I}}$ is the identity operator and

$$\hat{H} = \frac{1}{\sqrt{2}} \begin{bmatrix} 1 & 1 \\ 1 & -1 \end{bmatrix}, \tag{16}$$

is called the *Hadamard gate*. This transformation is equivalent to rotating the qubit $\sigma_0$ from the reference $\sigma_0^z$ basis the the $\sigma_0^x$ basis. A straightforward calculation leads to the following probability distribution of the projective measurement in the new basis $|\sigma_0^x, \sigma_1\rangle$:

$$P_b(\sigma_0^x, \sigma_1) = \frac{\Phi_{0\sigma_1}^2 + \Phi_{1\sigma_1}^2}{4} + \frac{1 - 2\sigma_0^x}{2} \Phi_{0\sigma_1} \Phi_{1\sigma_1} \cos(\Delta\theta), \tag{17}$$

where $\Delta\theta = \theta_{0\sigma_1} - \theta_{1\sigma_1}$. Therefore, the statistics collected by measuring in this basis implicitly contains partial information on the phases. To obtain the full phases structure, additional transformations are required, one example being the rotation from the reference basis to the $\sigma_j^y$ local basis, realized by the elementary gate

$$\hat{K} = \frac{1}{\sqrt{2}} \begin{bmatrix} 1 & -i \\ 1 & i \end{bmatrix}. \tag{18}$$

## 3.1 Setup

We now proceed to use QuCumber to reconstruct a complex-valued wavefunction. For simplicity, we restrict ourselves to two qubits and consider the general case of a quantum state with random amplitudes $\Phi_{\sigma_0\sigma_1}$ and random phases $\theta_{\sigma_0\sigma_1}$. This example is available in the online tutorial. [3] We begin by importing the required packages:

```
from qucumber.nn_states import ComplexWaveFunction
import qucumber.utils.unitaries as unitaries
import qucumber.utils.cplx as cplx
```

[3] The tutorial for complex wavefunctions can be found at https://qucumber.readthedocs.io/en/stable/_examples/Tutorial2_TrainComplexWaveFunction/tutorial_qubits.html

Since we are dealing with a complex wavefunction, we load the corresponding module ComplexWaveFunction to build the RBM quantum state $\psi_{\lambda\mu}(\boldsymbol{\sigma})$. Furthermore, the following additional utility modules are required: the utils.cplx backend for complex algebra, and the utils.unitaries module which contains a set of elementary local rotations. By default, the set of unitaries include local rotations to the $\sigma^x$ and $\sigma^y$ basis, implemented by the $\hat{H}$ and $\hat{K}$ gates respectively.

We continue by loading the data[4] into QuCumber, which is done using the load_data function of the data utility:

```
train_path = "qubits_train.txt"
train_bases_path = "qubits_train_bases.txt"
psi_path = "qubits_psi.txt"
bases_path = "qubits_bases.txt"

train_samples, true_psi, train_bases, bases = data.load_data(
    train_path, psi_path, train_bases_path, bases_path
)
```

As before, we may load the true target wavefunction from qubits_psi.txt, which can be used to calculate the fidelity and KL divergence. In contrast with the positive case, we now have measurements performed in different bases. Therefore, the training data consists of an array of qubits projections $(\sigma_0^{b_0}, \sigma_1^{b_1})$ in qubits_train_samples.txt, together with the corresponding bases $(b_0, b_1)$ where the measurement was taken, in qubits_train_bases.txt. Finally, QuCumber loads the set of all the bases appearing the in training dataset, stored in qubits_bases.txt. This is required to properly configure the various elementary unitary rotations that need to be applied to the RBM state during the training. For this example, we generated measurements in the following bases:

$$(b_0, b_1) = (z, z), (x, z), (z, x), (y, z), (z, y). \tag{19}$$

Finally, before the training, we initialize the set of unitary rotations and create the RBM state object. In the case of the provided dataset, the unitaries are the $\hat{H}$ and $\hat{K}$ gates. The required dictionary can be created with unitaries.create_dict(). By default, when unitaries.create_dict() is called, it will contain the identity, the $\hat{H}$ gate, and the $\hat{K}$ gate, with the keys Z, X, and Y, respectively. It is possible to add additional gates by specifying them as

```
U = torch.tensor([[<re_part>], [<im_part>]], dtype=torch.double)
unitary_dict = unitaries.create_dict(<unitary_name>=U)
```

where re_part, im_part, and unitary_name are to be specified by the user.

We then initialize the complex RBM object with

```
state = ComplexWaveFunction(
    num_visible=2, num_hidden=2, unitary_dict=unitary_dict
)
```

The key difference between positive and complex wavefunction reconstruction is the requirement of additional measurements in different basis. Despite this, loading the data, initializing models, and training the RBMs are all very similar to the positive case, as we now discuss.

---

[4] The training dataset can be downloaded from https://github.com/PIQuIL/QuCumber/blob/master/examples/Tutorial2_TrainComplexWaveFunction/

## 3.2 Training

Like in the case of a positive wavefunction, for the complex case QuCumber optimizes the network parameters to minimize the KL divergence between the data and the RBM distribution. When measuring in multiple bases, the optimization now runs over the set of parameters $(\lambda, \mu)$ and minimizes the sum of KL divergences between the data distribution $P(\sigma^b)$ and the RBM distribution $|\psi_{\lambda\mu}(\sigma^b)|^2$ for each bases $b$ appearing in the training dataset [6]. For example, if a given training sample is measured in the basis $(x, z)$, QuCumber applies the appropriate unitary rotation $\mathcal{U} = \hat{H}_0 \otimes \hat{\mathcal{I}}_1$ to the RBM state before collecting the gradient signal.

Similar to the case of positive wavefunction, we generate the Hilbert space (to compute fidelity and KL divergence) and initialize the callbacks

```
state.space = nn_state.generate_hilbert_space(2)
callbacks = [
    MetricEvaluator(
        log_every,
        {"Fidelity": ts.fidelity, "KL": ts.KL},
        target_psi=true_psi,
        bases=bases,
        verbose=True,
        space=state.space,
    )
]
```

The training is carried out by calling the `fit` function of `ComplexWaveFunction`, given the set of hyperparameters

```
state.fit(
    train_samples,
    epochs=100,
    pos_batch_size=10,
    neg_batch_size=10,
    lr=0.05,
    k=5,
    input_bases=train_bases,
    callbacks=callbacks,
)
```

In Fig. 5 we show the total KL divergence and the fidelity with the true two-qubit state during training. After successfully training a QuCumber model, we can once again compute expectation values of physical observables, as discussed in Section 2.3.

## 4 Conclusion

We have introduced the open source software package QuCumber, a quantum calculator used for many-body eigenstate reconstruction. QuCumber is capable of taking input data representing projective measurements of a quantum wavefunction, and reconstructing the wavefunction using a restricted Boltzmann machine (RBM).

Once properly trained, QuCumber can produce a new set of measurements, sampled stochastically from the RBM. These samples, generated in the reference basis, can be used to verify the training of the RBM against the original data set. More importantly, they can be used to calculate expectation values of many physical observables. In fact, any expectation value typically estimated by conventional Monte Carlo methods can be implemented as an estimator in QuCumber. Such estimators may be inaccessible in the reference basis, for example. Or, they

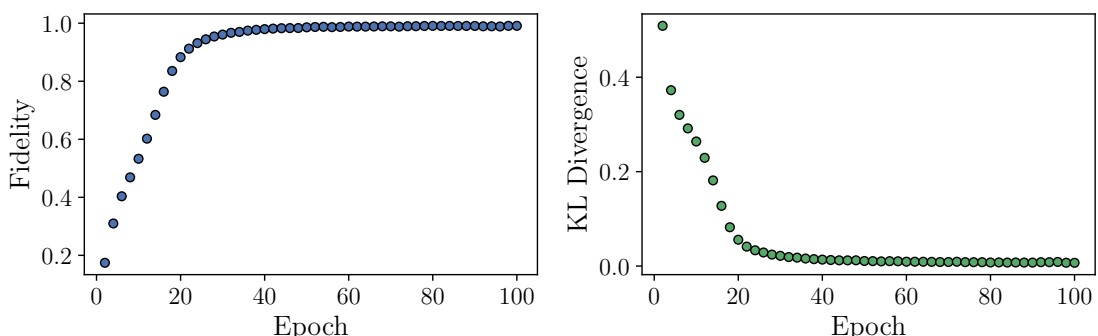

Figure 5: Training a complex RBM with QuCumber on random two-qubit data. We show the fidelity (left), and KL divergence (right), as a function of the training epochs.

may be difficult or impossible to implement in the setup for which the original data was obtained. This is particularly relevant for experiments, where it is easy to imagine many possible observables that are inaccessible, due to fundamental or technical challenges.

Future versions of QuCumber, as well as the next generation of quantum state reconstruction software, may explore different generative models, such as variational autoencoders, generative adversarial networks, or recurrent neural networks. The techniques described in this paper can also be extended to reconstruct mixed states, via the purification technique described in Reference [7]. In addition, future techniques may include hybridization between machine learning and other well-established methods in computational quantum many-body physics, such as variational Monte Carlo and tensor networks [8].

# Acknowledgements

We acknowledge M. Albergo, G. Carleo, J. Carrasquilla, D. Sehayek, and L. Hayward Sierens for stimulating discussions. We thank the Perimeter Institute for Theoretical Physics for the continuing support of PIQuIL.

**Author contributions**   Authors are listed alphabetically. For an updated record of individual contributions, consult the repository at https://github.com/PIQuIL/QuCumber/graphs/contributors.

**Funding information**   This research was supported by the Natural Sciences and Engineering Research Council of Canada (NSERC), the Canada Research Chair program, and the Perimeter Institute for Theoretical Physics. We also gratefully acknowledge the support of NVIDIA Corporation with the donation of the Titan Xp GPU used in this work. Research at Perimeter Institute is supported in part by the Government of Canada through the Department of Innovation, Science and Economic Development Canada and by the Province of Ontario through the Ministry of Economic Development, Job Creation and Trade. P. H. acknowledges support from ICFOstepstone, funded by the Marie Sklodowska-Curie Co-funding of regional, national and international programmes (GA665884) of the European Commission, as well as by the Severo Ochoa 2016–2019' program at ICFO (SEV–2015–0522), funded by the Spanish Ministry of Economy, Industry, and Competitiveness (MINECO).

## A  Glossary

This section contains an overview of terms discussed in the document which are relevant for RBMs. For more detail we refer the reader to the code documentation on https://qucumber. readthedocs.io/en/stable/, and References [12,30].

- *Batch*: A subset of data upon which the gradient is computed and the network parameters are adjusted accordingly. A smaller batch size often results in a more stochastic trajectory, while a large batch size more closely approximates the exact gradient and has less variance.

- *Biases*: Adjustable parameters in an RBM, denoted by $b_j$ and $c_i$ in Eq. (20).

- *Contrastive divergence*: An approximate maximum-likelihood learning algorithm for RBMs [12]. CD estimates the gradient of the effective energy (20) with respect to model parameters by using Gibbs sampling to compare the generated and target distributions.

- *Energy*: The energy of the joint configuration $(\boldsymbol{v}, \boldsymbol{h})$ of a RBM is defined as follows:

$$E_\lambda(\boldsymbol{v}, \boldsymbol{h}) = -\sum_{j=1}^{n_v} b_j v_j - \sum_{i=1}^{n_h} c_i h_i - \sum_{ij} h_i W_{ij} v_j. \tag{20}$$

- *Effective energy*: Obtained from the energy by tracing out the hidden units $\boldsymbol{h}$; often called the "free energy" in machine learning literature.

$$\mathcal{E}_\lambda(\boldsymbol{v}) = -\sum_{j=1}^{n_v} b_j v_j - \sum_{i=1}^{n_h} \log \left[ 1 + \exp\left( \sum_{j}^{n_v} W_{ij} v_j + c_i \right) \right]. \tag{21}$$

- *Epoch*: A single pass through an entire training set. For example, with a training set of 1,000 samples and a batch size of 100, one epoch consists of 10 updates of the parameters.

- *Gibbs sampling*: A Monte Carlo algorithm that samples from the conditional distribution of one variable, given the state of other variables. In an RBM, the restricted weight connectivity allows Gibbs sampling between the visible "block", conditioned on the hidden "block", and vice versa.

- *Hidden units*: There are $n_h$ units in the hidden layer of the RBM, denoted by the vector $\boldsymbol{h} = (h_1, \ldots, h_{n_h})$. The number of hidden units can be adjusted to increase the representational capacity of the RBM.

- *Hyperparameters:* A set of parameters that are not adjusted by a neural network during training. Examples include the learning rate, number of hidden units, batch size, and number of training epochs.

- *Joint distribution*: The RBM assigns a probability to each joint configuration $(\boldsymbol{v}, \boldsymbol{h})$ according to the Boltzmann distribution:

$$p_\lambda(\boldsymbol{v}, \boldsymbol{h}) = \frac{1}{Z_\lambda} e^{-E_\lambda(\boldsymbol{v}, \boldsymbol{h})}. \tag{22}$$

- *KL divergence*: The Kullback-Leibler divergence, or relative entropy, is a measure of the "distance" between two probability distributions $P$ and $Q$, defined as:

$$\mathrm{KL}(P \,\|\, Q) = \sum_{\boldsymbol{v}} P(\boldsymbol{v}) \log \frac{P(\boldsymbol{v})}{Q(\boldsymbol{v})}. \tag{23}$$

  The KL divergence between two identical distributions is zero. Note that it is not symmetric between $P$ and $Q$.

- *Learning rate*: The step size used in the gradient descent algorithm for the optimization of the network parameters. A small learning rate may result in better optimization but will take more time to converge. If the learning rate is too high, training might not converge or will find a poor optimum.

- *Marginal distribution*: Obtained by marginalizing out the hidden layer from the joint distribution via

$$p_{\lambda}(\boldsymbol{v}) = \frac{1}{Z_{\lambda}} \sum_{\boldsymbol{h}} e^{-E_{\lambda}(\boldsymbol{v}, \boldsymbol{h})} = \frac{1}{Z_{\lambda}} e^{-\mathcal{E}_{\lambda}(\boldsymbol{v})}. \tag{24}$$

- *QuCumber*: A quantum calculator used for many-body eigenstate reconstruction.

- *Parameters*: The set of weights and biases $\boldsymbol{\lambda} = \{\boldsymbol{W}, \boldsymbol{b}, \boldsymbol{c}\}$ characterizing the RBM energy function. These are adjusted during training.

- *Partition function*: The normalizing constant of the Boltzmann distribution. It is obtained by tracing over all possible pairs of visible and hidden vectors:

$$Z_{\lambda} = \sum_{\boldsymbol{v}, \boldsymbol{h}} e^{-E_{\lambda}(\boldsymbol{v}, \boldsymbol{h})}. \tag{25}$$

- *Restricted Boltzmann Machine*: A two-layer network with bidirectionally connected stochastic processing units. "Restricted" refers to the connections (or weights) between the visible and hidden units: each visible unit is connected with each hidden unit, but there are no intra-layer connections.

- *Visible units*: There are $n_v$ units in the visible layer of the RBM, denoted by the vector $\boldsymbol{v} = (v_1, \ldots, v_{n_v})$. These units correspond to the physical degrees of freedom. In the cases considered in this paper, the number of visible units is equal to the number of spins $N$.

- *Weights*: $W_{ij}$ is the symmetric connection or interaction between the visible unit $v_j$ and the hidden unit $h_i$.

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
