# Peer review of "QuCumber: wavefunction reconstruction with neural networks"

_SciPost Physics, doi:SciPost Phys. 7, 009 (2019)_

## Round 1 · Referee Report · Anonymous (Referee 2) · 2019-2-6

Strengths

  1. Very interesting research avenue and methodology.
  2. The open source software will be very useful

Weaknesses

  1. The mathematical problem solved is insufficiently defined
  2. No novelty besides the illustration of the software package QuCumber. This is not necessarily an issue, but the manuscript is not a traditional research paper.

Report

The authors present the software QuCumber which aims at reconstructing a wavefunction from
observable mutli-qubits data. The software uses a Restricted Boltzman Machine to construct the trial wavefunction. This is high level research in a very interesting topic. I think the manuscript should be published provided the type of article (illustration of a software) is compatible with Scipost editorial policy. I am personally strongly in favour of the associated open research approach. Before being published however, the authors must add some defining material, as outlined below, to make the article self-contained. I also have two physics questions for the authors.

The point of quantum computers is that they are (supposed to be) very difficult to simulate classically because of the exponential size of the Hilbert space. By using a RBM state (which is not exponential), don’t the authors implicitly assume that the state has a much simpler structure? Doesn’t that defy the usefulness of quantum computers or of the RBM for this problem?
A related questions is that actual qubits don’t generate pure states but rather a statistical mixture of
states. Will QuCumber be useful for actual experimental data?

Requested changes

  1. Define precisely (with equations not words) the mathematical problem solved by QuCumber. What is the RBM trial wavefuntion? What is the cost function which is optimized?

  2. Add a short discussion of the applications of QuCumber (see the physics questions above).

  3. Define the entanglement entropy. Define its relation to the SWAP operator. It is also a bit strange that no actual data is shown for this quantity.

  4. In the introduction, clearly list the articles that explain QuCumber theory as well as those where QuCumber has been used to solve a physics problem.

  • validity: top
  • significance: high
  • originality: high
  • clarity: good
  • formatting: excellent
  • grammar: excellent

Author:  Roger Melko  on 2019-05-17  [id 516]

(in reply to Report 2 on 2019-02-06)
Category:
answer to question
correction

As pointed out by the referee, the point of quantum computers is that they are exponentially difficult to simulate classically. However, given a quantum computer that prepares a wavefunction of some "structure" (phase or entanglement structure, say) it is not immediately obvious whether the difficulty of numerical reconstruction from projective measurement data is also necessarily scales exponential. The RBM itself doesn't assume any locality or dimensionality; it is known for example to faithfully represent volume-law wave functions. They therefore may be useful for states with arbitrary structure (although much further work is necessary to study this).

Second: yes, actual qubits will generally produce statistical mixtures of states. Generalizations of our RBM procedures, like the purification described in Torlai and Melko, "Latent space purification via neural density operators", Phys. Rev. Lett. 120, 240503 (2018), can be used in this case on real experimental data. This point has now been added to the last paragraph of the Conclusions.

Response to report 2 "Requested Changes"

  1. Define precisely (with equations not words) the mathematical problem solved by QuCumber. What is the RBM trial wavefuntion? What is the cost function which is optimized?

The mathematical definitions are described in a precise and self-contained way, largely in the Glossary. The trial wavefunction (the marginal distribution) and the cost function (KL divergence) are defined in equation (24) and (23) respectively. We have now referred to these equations in the main text for increased clarity and readability.

  1. Add a short discussion of the applications of QuCumber (see the physics questions above).

We have added to the introduction and conclusion in this regard.

  1. Define the entanglement entropy. Define its relation to the SWAP operator. It is also a bit strange that no actual data is shown for this quantity.

We have defined the relative quantities and included a new plot in section 2.3.3

  1. In the introduction, clearly list the articles that explain QuCumber theory as well as those where QuCumber has been used to solve a physics problem.

QuCumber theory is almost completely self contained in References 5 and 6, now clearly cited in the last paragraph of the introduction. In addition, we have cited the new review [28] in the introduction which contains an extensive review of RBM theory and applications quantum physics problems.

---

## Round 1 · Referee Report · Everard van Nieuwenburg (Referee 1) · 2019-2-6

Strengths

1- Clearly written & has example code snippets 2-Concise bits of theory explaining the background in between the snippets 3-Associated code with extensive tutorials available online

Weaknesses

1- I missed a bit of discussion on/intuition for the constraints of the package

Report

The creatively named QuCumber package (part of PIQuIL at the Perimeter Institute) provides a python implementation for performing quantum state tomography on experimental and numerical data. To do so, it uses Restricted Boltzmann Machines (RBM) as an Ansatz for pure-state wavefunctions and optimizes them using standard RBM training techniques. The premise for releasing such a package is that current-day experimental systems are at a level where 1) data-sets for tomography in this approach can be generated and 2) tomography methods become essential in verifying experiments where classical simulations no longer suffice.

Previous works, cited in the article, have shown (emperically) that RBM-based wavefunctions provide a powerful Ansatz. Having a streamlined python package that facilitates tomography with RBMs without the user having to implement training procedures is extremely helpful, and I expect that a package like this may well become a standard tool.

A remark should be made w.r.t. the NetKet package, which supports quantum state tomography with RBMs in exactly the same way. An advantage of QuCumber, at least currently, is that it is based on the pyTorch backend and hence fully transparently supports running the algorithm on a GPU. The package has potential to be extended and include other generative models easily, too.

There are some minor things in the paper left unexplained, which in light of the pedagogical nature of the rest of the paper could use a small addition. For example, the paper mentions (block) Gibbs sampling (and e.g. the positive/negative phases of the contrastive divergence algorithm), but the block sampling procedure is never mentioned in the Glossary (although to be fair, a more extensive explanation of the theory can be found in the associated github repository). In the code snipped split between pages 4&5 the code requires the generation of a Hilbert space description as an argument to the MetricEvaluator, which is never referenced in the text.

A discussion paragraph on the applicability of the package in practical terms would be very useful for prospective users. Do the authors have any intuition regarding the scaling of the required number of samples (to train and also to estimate reconstruction fidelities from measurements/energy), for example? And how successful is the 2-RBM setup for complex wavefunctions in general? These points are discussed in more detail on the associated github repository, but would be helpful in the paper.

Last, out of curiosity, I would like to ask if the authors could use the approach to train e.g. an RBM with the magnetic field as a conditional parameter (c.f. Fig. 2), so that one machine suffices?

Overall the submission introduces a numerical package based on previous results. The main aim of the submission is to introduce this package through example snippets and demonstrate it's capabilities, whilst further details are left to the associated open source github repository. SciPost matches the open source nature of this project, and can reach the experimental (and numerical) communities that may benefit from this package. I have no reservations therefore in considering this submission suitable for SciPost.

Requested changes

1- Add a small snippet on Gibbs sampling (in Glossary)

2- Typo on page 5: we have demonstrated to -> the

  • validity: top
  • significance: top
  • originality: high
  • clarity: top
  • formatting: excellent
  • grammar: perfect

Author:  Roger Melko  on 2019-05-17  [id 517]

(in reply to Report 1 by Everard van Nieuwenburg on 2019-02-06)

We thanks the referee for his thoughtful report.

Response to report 1 "Requested Changes"

1- Add a small snippet on Gibbs sampling (in Glossary)

This has been added.

2- Typo on page 5: we have demonstrated to -> the

Done.

---

## Round 2 · Referee Report · Everard van Nieuwenburg (Referee 1) · 2019-5-19

Report

The authors have added the pieces I thought would make the submission more complete, and I remain with my conclusion that it is suitable and ready for SciPost.

---

## Round 2 · Referee Report · Anonymous (Referee 2) · 2019-5-20

Report

The manuscript can be published in its current form.

---

## Round 2 · List of Changes

Response to report 1 "Requested Changes"

1- Add a small snippet on Gibbs sampling (in Glossary)

This has been added.

2- Typo on page 5: we have demonstrated to -> the

Done.

Response to report 2 "Requested Changes"

  1. Define precisely (with equations not words) the mathematical problem solved by QuCumber. What is the RBM trial wavefuntion? What is the cost function which is optimized?

The mathematical definitions are described in a precise and self-contained way, largely in the Glossary. The trial wavefunction (the marginal distribution) and the cost function (KL divergence) are defined in equation (24) and (23) respectively. We have now referred to these equations in the main text for increased clarity and readability.

  1. Add a short discussion of the applications of QuCumber (see the physics questions above).

We have added to the introduction and conclusion in this regard.

  1. Define the entanglement entropy. Define its relation to the SWAP operator. It is also a bit strange that no actual data is shown for this quantity.

We have defined the relative quantities and included a new plot in section 2.3.3

  1. In the introduction, clearly list the articles that explain QuCumber theory as well as those where QuCumber has been used to solve a physics problem.

QuCumber theory is almost completely self contained in References 5 and 6, now clearly cited in the last paragraph of the introduction. In addition, we have cited the new review [28] in the introduction which contains an extensive review of RBM theory and applications quantum physics problems.

---

## Editorial Decision

published